# Springtime winds drive Ross Sea ice variability and change in the following autumn

Marika M. Holland[1], Laura Landrum [1], Marilyn Raphael [2] & Sharon Stammerjohn[3]

Autumn sea ice trends in the western Ross Sea dominate increases in Antarctic sea ice and are outside the range simulated by climate models. Here we use a number of independent data sets to show that variability in western Ross Sea autumn ice conditions is largely driven by springtime zonal winds in the high latitude South Pacific, with a lead-time of 5 months. Enhanced zonal winds dynamically thin the ice, allowing an earlier melt out, enhanced solar absorption, and reduced ice cover the next autumn. This seasonal lag relationship has implications for sea ice prediction. Given a weakening trend in springtime zonal winds, this lagged relationship can also explain an important fraction of the observed sea ice increase. An analysis of climate models indicates that they simulate weaker relationships and wind trends than observed. This contributes to weak western Ross Sea ice trends in climate model simulations.

[1] National Center for Atmospheric Research, P.O. Box 3000, Boulder, CO 80307-3000, USA. [2] Department of Geography, University of California, Los Angeles, 1255 Bunche Hall, Los Angeles, CA 90095-1524, USA. [3] Institute of Arctic and Alpine Research, University of Colorado, Campus Box 450, Boulder, CO 80309-0450, USA. Correspondence and requests for materials should be addressed to M.M.H. (email: mholland@ucar.edu)

Observed Southern Hemisphere sea ice extent has increased since 1979[1] in contrast to results from most global climate models[2, 3]. The observed trend is made up of large regional changes that are partly compensating. While previous work has indicated that the observed increases in the total Antarctic sea ice extent are consistent with natural variability simulated by climate models[4], this is not the case if the regional and seasonal nature of the trends is considered.

Of particular interest is the western Ross Sea region during the autumn ice advance season. This is the region and season of the largest ice increases in the Southern Hemisphere (Fig. 1a, b). These observed regional trends are larger than those simulated by a group of coupled climate models for the 20th century, making them difficult to reconcile with simulated internal variability[5]. Additionally, the western Ross Sea ice trends do not appear attributable to observed changes in many modes of variability[6], including the significant trends in the austral summer Southern Annular Mode[7], which are associated with stratospheric ozone loss. A number of previous studies[8–11] have identified mechanisms that can cause increased sea ice around Antarctica. However, these generally do not explain the increases observed in the western Ross Sea during autumn.

Trends in the Ross Sea ice extent have been associated with coincident wind-driven ice transport anomalies[12]. Observational evidence has also indicated that seasonal-lagged relationships influence ice trends. In particular, ice concentration trends in autumn have been related to changes in ice during the previous spring season[13]. The presence of seasonal memory in Antarctic sea ice-ocean interactions has also been documented in the context of interannual variability[14–16], with ice conditions during the autumn ice advance season significantly related to sea ice during the previous spring ice retreat.

Here we use a number of independent data sets to assess factors that contribute to variability and change in western Ross Sea ice cover during autumn. This includes the analysis of mechanisms driving a seasonal memory of the sea ice and the possible consequences for observed sea ice trends in the western Ross Sea. We show that, on interannual timescales, wind variability in the previous October is a strong predictor of ice anomalies in March to May, with implications for seasonal ice predictability. On the basis of these seasonal-lagged relationships, we find that trends in springtime winds could account for up to 30% of the autumn sea ice trends in the western Ross Sea. Climate models simulate weaker relationships and wind trends than observed, which

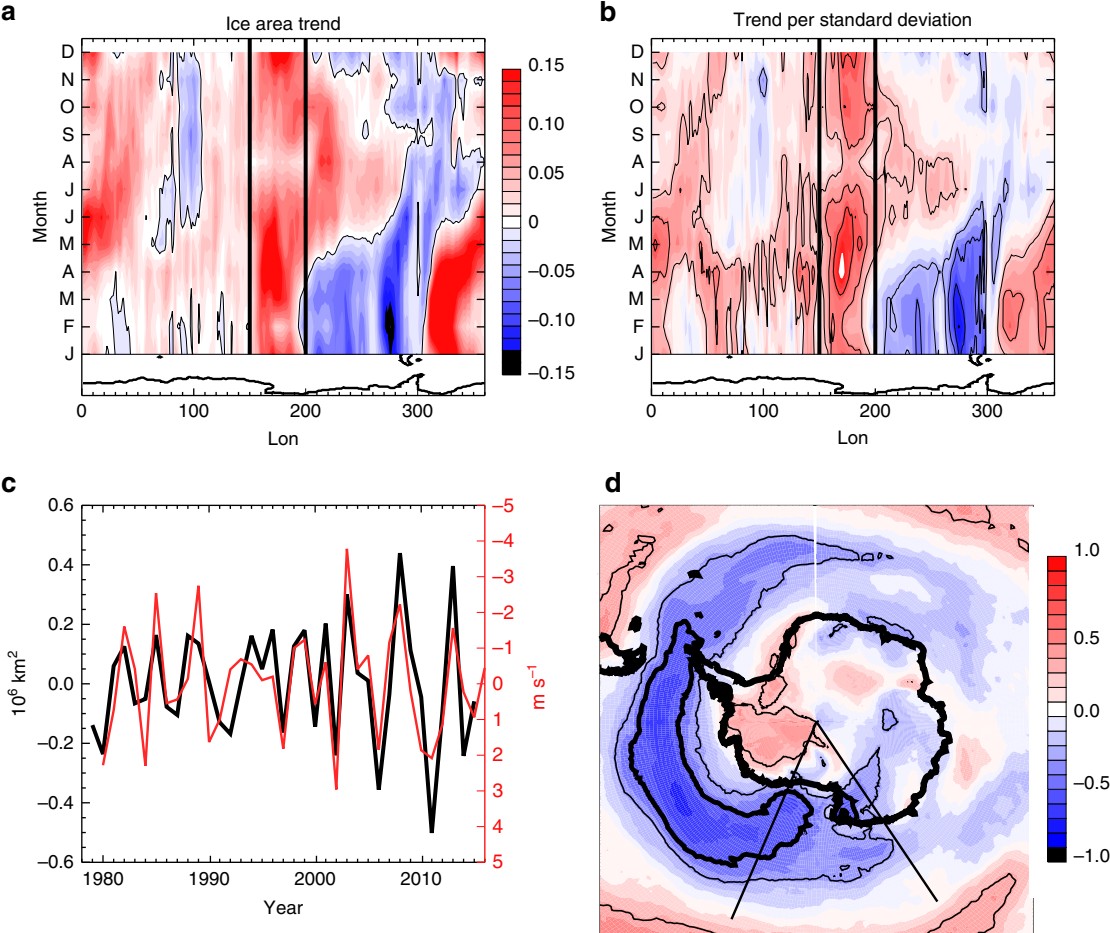

**Fig. 1** Observed sea ice properties and relationship to October zonal winds. **a** Sea ice area trend in 10⁶ km² per 37 years, from 1979 to 2015 as a function of longitude (at every 1.125°) and month. **b** Sea ice area trend relative to the interannual standard deviation as a function of longitude and month. The *lined contour* interval is 1 standard deviation. The zero contour is omitted. **c** The timeseries of March sea ice area anomalies in the 150–200° E region (*black; left* axis) and the averaged preceding October zonal wind anomaly index (*red; right* axis). The two timeseries have been detrended. **d** The correlation of the preceding gridded October zonal wind anomalies with the March western Ross Sea ice area. The *thin lined contour* indicates the 95% significance level. The zonal wind index timeseries shown in **c** is the average over the region of high negative correlation (R < −0.5) that is indicated by the *bold black contour* on **d**. The 150–200° E region is indicated on **d** by the *thin black lines*

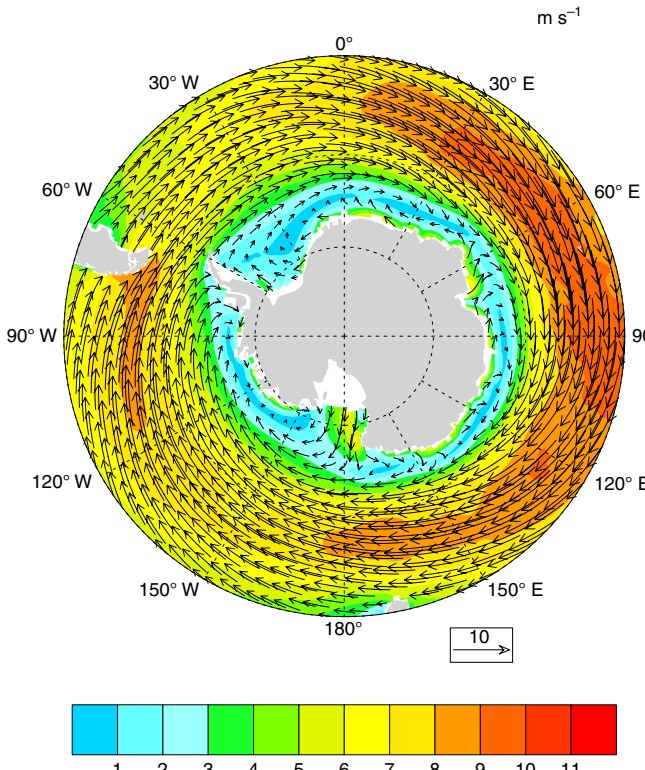

**Fig. 2** October climatological winds. The October climatological winds averaged from 1979 to 2015 from ERA-Interim. *Colored shading* indicates the wind magnitude in meters per second

influences the magnitude of western Ross Sea ice trends that they simulate.

## Results

**Relationship between sea ice variability and winds**. We assess 1979–2015 autumn conditions in the western Ross Sea, defined as the region between 150 and 200º E, as this is the area with the largest trends in an absolute sense and also relative to the inter-annual variability (Fig. 1a, b). The long-term March-May average sea ice trends in this region are about 60% of the total Antarctic autumn ice area trend. Fig. 1c shows the timeseries of satellite-derived detrended western Ross Sea ice area anomalies in March. The ice anomalies in March persist for several months, with associated very high and significant correlation to ice area from April to June ($R = 0.85, 0.74, 0.65$ for AMJ, respectively). This suggests that much of the ice variability in autumn in the western Ross Sea is associated with variations in the ice advance beginning in March.

Previous studies have highlighted the importance of wind variations for sea ice drift and concentration[12]. Correlation analysis using the ERA-Interim data indeed suggests that the western Ross Sea March ice area anomalies are strongly related to wind variability. However, somewhat counter to expectations, the ice area anomalies only show minimal correlation to the simultaneous March or prior February winds (Supplementary Fig. 1). Instead the March ice area anomalies are most strongly associated with winds 5 months earlier in October (Fig. 1d). The correlations are particularly high for the zonal wind anomalies, with decreased March ice cover associated with stronger October zonal winds over the high latitude South Pacific, a region where westerly winds dominate the climatology (Fig. 2).

To quantify the October zonal winds that are important for western Ross Sea ice conditions, we create a wind index by

averaging the zonal wind over the region of high negative correlation ($R < -0.5$ as shown on Fig. 1d). At this 5-month lead time, this regional zonal wind variability, shown on Fig. 1c, is significantly correlated to the detrended March ice variability at $R = -0.75$. As shown in Supplementary Fig. 2, this results from significantly correlated reductions in March sea ice throughout the western Ross Sea domain. This correlation is considerably higher than the relationship of March sea ice area in the western Ross Sea to winds in any other month (Supplementary Fig. 1).

Sea ice area correlations are stronger with the zonal wind than with other atmospheric circulation metrics (Supplementary Figs. 3 and 4). However this zonal wind variability is associated with the large-scale atmospheric circulation, with associated anomalies in the meridional wind and sea level pressure. In particular, the October zonal wind timeseries is significantly correlated ($R = -0.70$) with variability in the depth of the October Amundsen Sea Low (ASL), which is a climatological low pressure center in the region[17, 18]. Many modes of variability such as the El Niño-Southern Oscillation[17, 19] (and associated Pacific–South American pattern), the Southern Annular Mode[17, 19], the Pacific Decadal Oscillation[11] and the Atlantic Multi-decadal Oscillation[20], influence the ASL, providing links to tropical and global variability. The correlation of detrended western Ross Sea ice and the October ASL is $R = 0.50$, meaning that a deeper October ASL is associated with less March ice cover. This is compared to a correlation of $R = -0.75$ for the zonal wind index. Interestingly, the relationship between March sea ice area in the western Ross Sea and the depth of the ASL is of opposite sign for the seasonal-lagged relationships discussed here compared to a contempora-neous relationship[21]. The fact that the zonal wind shows stronger relationships to western Ross Sea ice area than the ASL pressure alone suggests that various factors such as the location of the ASL also may play a role. This is generally consistent with other work assessing concurrent relationships that has indicated that not only the depth of the ASL but also its location can affect relationships to sea ice[22].

Variations in the October winds drive changes in sea ice motion and atmospheric heat transport. Both of these can affect changes in sea ice area. However, relationships to atmospheric temperature (Supplementary Fig. 5), which could indicate atmospheric heat transport anomalies, are not of a consistent sign for stronger winds to drive ice loss in the western Ross Sea. This suggests that wind-driven ice motion changes dominate the seasonal-lagged effects on ice area. We diagnose the relationship of winds to sea ice motion using the monthly averaged Polar Pathfinder data[23]. As indicated by the regression of October ice motion on the October zonal wind index (Fig. 3), increased zonal winds drive enhanced ice transport from the western Ross Sea. Using ice area divergence diagnosed from the ice motion data and sea ice concentration, we find that stronger zonal winds are highly related to an enhanced net dynamical ice area loss from the western Ross Sea. More specifically, the net October mean ice area divergence from the western Ross Sea, which is equivalent to the net ice area transport from the region, is significantly correlated to the zonal wind anomalies at $R = -0.54$.

This dynamical ice loss should lead to a thinning of sea ice in the western Ross Sea, which in turn allows for an earlier melt out of sea ice, and an earlier ice retreat. As shown in Fig. 3, the relationship of the October zonal wind index to the timing of ice retreat is strongest in the southern-most part of the sea ice domain, where the relationship is highly significant. With this earlier ice retreat, a longer ice-free season results, allowing for enhanced shortwave absorption in the ocean over the summer months (Fig. 4). As indicated by the regression of sea surface temperature (SST) on the zonal wind anomalies (Fig. 4b), this results in an anomalously warm surface ocean over the summer

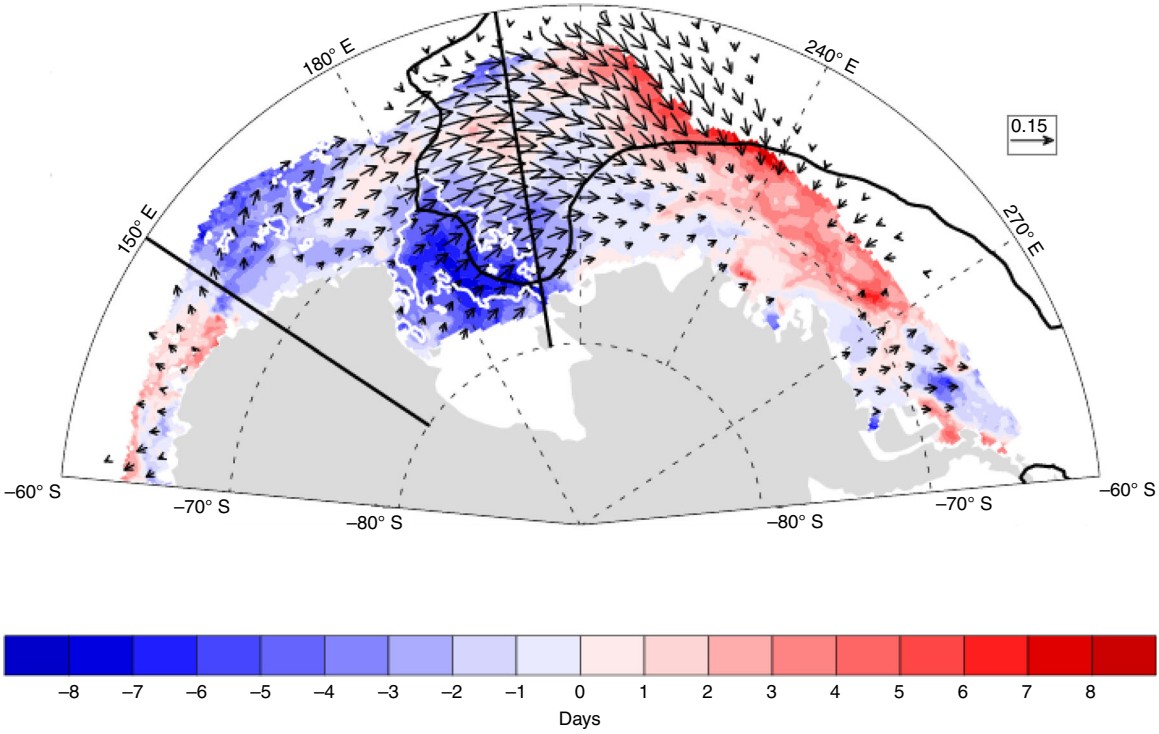

**Fig. 3** Relationship of ice motion and ice retreat to the October zonal winds. Vectors indicate the regression of October ice motion on the October zonal wind anomaly index (from Fig. 1c). *Colored shading* indicates the timing of ice retreat regressed on the normalized October zonal wind anomaly index in units of days. The *white contour* indicates where negative regression values are significant at the 95% level. The *black contour* indicates the −0.5 correlation of gridded October zonal winds and March Western Ross Sea ice area that is used to define the wind index (as shown on Fig. 1c). Vectors with a magnitude of < 0.02 are not shown

months, which delays ice advance (Fig. 4a) and results in lower sea ice area in March and subsequent months.

Previous studies have indicated that ice advance conditions in the autumn are correlated to previous spring ice retreat conditions[14–16]. Here we have related this to wind variations in the springtime and illustrated a mechanism, supported by a number of independent observationally based data sets, that provides this seasonal lag. This mechanism appears to be very important for driving interannual variations in autumn sea ice area in the western Ross Sea. However, the question remains to what extent this might explain the considerable sea ice trends in this region where sea ice is advancing earlier and retreating later[14]. Notably the study by Holland[13] provides some indication that the trends in March ice area are also associated with processes occurring during the previous spring. Here we further quantify whether the interannual relationship of October winds and March sea ice might explain some fraction of the sea ice area trends.

**Attribution of the observed autumn sea ice trend.** Figure 5 shows the trends in gridded October zonal winds for 1979–2015 from ERA-Interim. The correlation of October zonal winds and western Ross Sea March ice area (also shown in Fig. 1d) is overlaid to highlight the relevant regions of interest. There are sizable decreases in October zonal winds in the Ross Sea region, which are also evident in JRA-55 and MERRA reanalysis products (Supplementary Fig. 6). The region of these wind decreases overlaps with and extends a bit west from the region of the highest interannual correlations. If we average the winds over the region with relatively large trends (*black area* in Fig. 5b), we find that the correlation to the subsequent March sea ice remains quite high at $R = −0.60$ (Fig. 5c). These winds are most influential though in the western part of the western Ross Sea ice analysis domain (Supplementary Fig. 1b). Notably, the March ice area

trends are also a maximum in the western part of the ice domain. Other properties, such as the ice motion, timing of ice advance and retreat, and shortwave absorption anomalies also have significant correlations to the October zonal winds in the high trend region (Supplementary Figs. 7 and 8), suggesting that similar physical processes as those described above link these zonal wind anomalies to sea ice conditions 5 months later.

To quantify the role that the wind trends play for sea ice, we use a regression analysis to diagnose the change in western Ross Sea ice area associated with the wind variability on interannual timescales. Multiplying this by the relevant wind trend then provides a measure of the change in sea ice area associated with the changing winds (Fig. 5d). This assumes that the linear relationship obtained through the regression analysis is applicable to longer timescales. There is some reason to be skeptical about this assumption given that recent work has indicated a two-timescale response of sea ice to wind variations[24] in which the short term (interannual) response can differ markedly from the longer term (decadal) response due to the influence of ocean circulation changes. Nevertheless, if we make these assumptions (and based on the model analysis discussed below, there is some reason to believe them), we find that the trends in the zonal winds in October can account for 20–30% of the trend in the western Ross Sea ice area in March and April (Fig. 5d). While this indicates that other processes are also involved, it does suggest that trends in the October zonal winds may play an important role for the autumn sea ice trend in the western Ross Sea. Notably, other factors have been implicated for explaining sea ice trends in different regions and seasons[8–11], indicating that low-frequency Antarctic sea ice variability and trends are influenced by multiple processes.

**Simulation of sea ice and winds in climate models.** As noted above, climate models do not simulate austral fall western Ross Sea ice trends of the magnitude that have been observed[5]. We

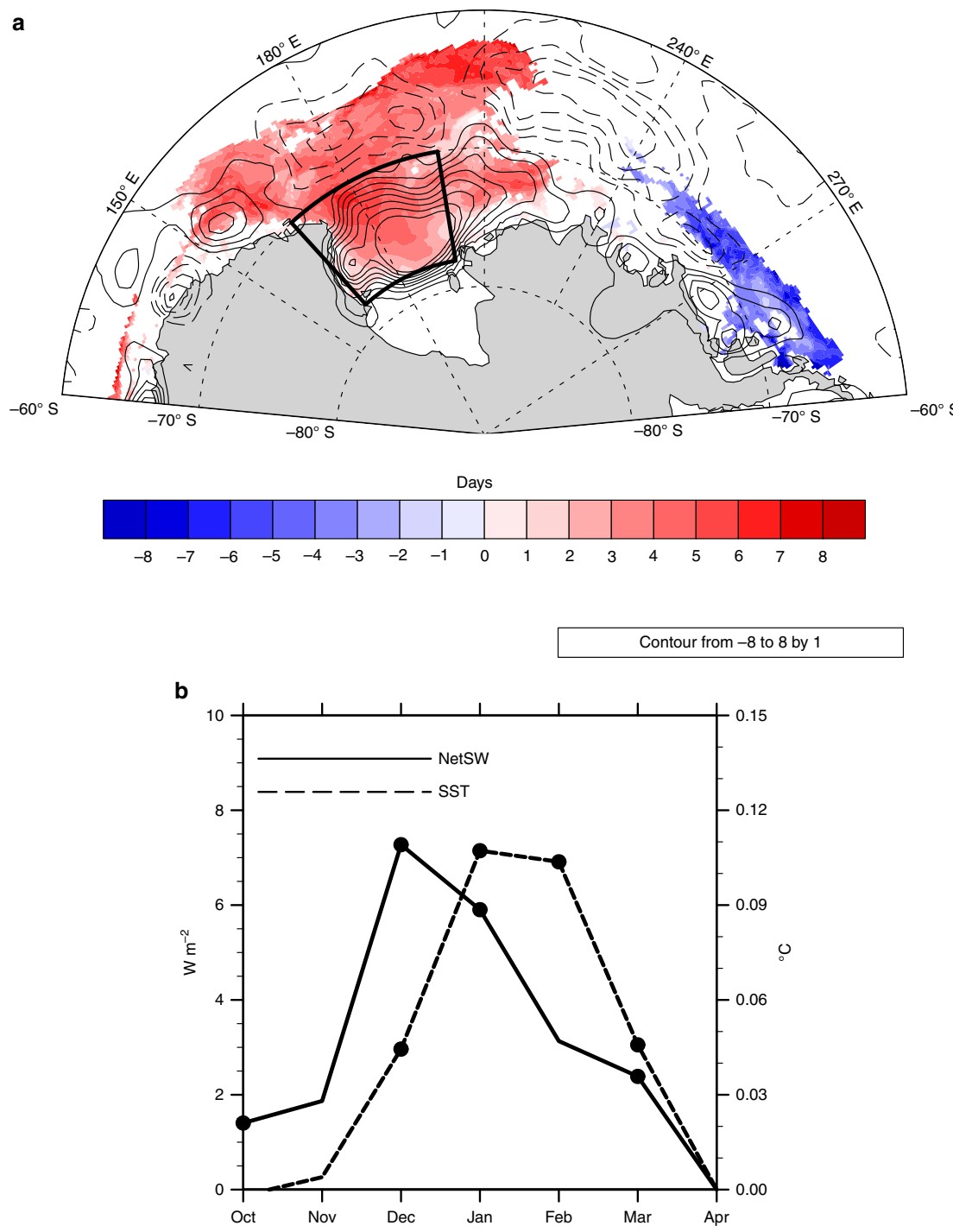

**Fig. 4** The relationship of environmental properties to the preceding October zonal winds. **a** Regression of 1979–2015 surface net shortwave radiation averaged from December to February (*black contours*) and the timing of ice advance (*colored shading*) on the normalized and detrended prior October zonal wind index. The net shortwave regression is contoured at 1 W m$^{-2}$, negative values are *dashed*, and the zero line omitted. Regression of ice advance is only shown where significant at the 95% level. **b** Regression of monthly 1979–2015 net shortwave (*solid; left* axis) and 1982–2015 SST (*dash; right* axis) averaged in the region denoted by the *box* on **a** on the normalized and detrended October zonal wind index. Significant values are denoted by the *circles*

assess whether this may in part be associated with their simulation of wind influences on western Ross Sea ice variations, by analyzing over 11,000 years of output from 20 models participating in the Coupled Model Intercomparison Project 5 (CMIP5; Supplementary Table 1). We use output from pre-industrial control simulations given that there is little evidence that anthropogenic forcing from ozone loss or greenhouse gases are

driving Antarctic wind changes in spring[25, 26]. We consider that models could be potentially deficient in their simulation of the seasonal-lagged influence of October winds on autumn sea ice area, their simulation of October wind trends, or a combination of the two.

To assess the model relationships between winds and sea ice, we compute the correlation of October zonal winds averaged over

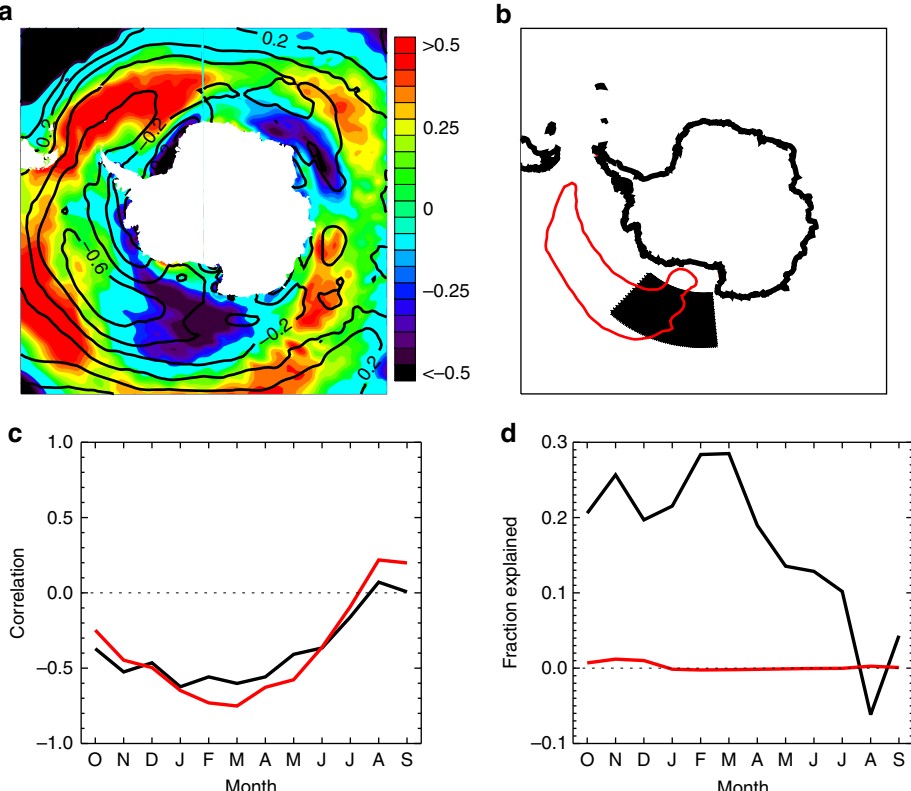

**Fig. 5** Attribution of western Ross Sea ice area trends to the October zonal winds trends. **a** Trends in gridded October zonal winds for 1979–2015 from ERA-Interim in the *colored shading* in units of m s$^{-1}$ per decade. *Black contours* indicate the correlation of the October zonal wind index and March western Ross Sea ice area (as also shown in Fig. 1d). **b** The two regions used for averaging the zonal wind anomalies. **c** The correlation of monthly western Ross Sea ice area and October zonal wind anomalies averaged for the two regions shown on **b**. **d** The fraction of the observed monthly trend in western Ross Sea ice area that is attributable to the October zonal wind trend averaged for the two regions. In **c** and **d**, the *color of the line* corresponds to the use of wind data that is averaged for the associated *colored region* in **b**

the *black region* shown on Fig. 5b with the following March western Ross Sea ice area in the models and observations. To enable a consistent comparison with observations, we consider these correlations for every possible 37-year timeseries from the models. Figure 6a shows the distribution of correlation coefficients obtained in this analysis. As in observations, the models typically simulate a negative correlation, indicating that anomalously weak October westerly winds are associated with increased western Ross Sea ice area 5 months later. However, the models only rarely (<2% of the time) simulate a correlation as strong as observed. Additionally, for 22% of the 37-year instances from the models, the models simulate a positive correlation in contrast to observations.

Some models are more similar to observations and exhibit reasonably high correlations (Supplementary Table 1). These models do indicate that increasing western Ross Sea March ice area is associated with weakening zonal winds on multi-decadal timescales. For example, from a long pre-industrial control integration of the Community Earth System Model Large Ensemble model[27], the correlation between all 37-year trends in the October zonal winds and 37-year trends in the western Ross Sea ice area is $R = -0.57$ (Supplementary Fig. 9). This suggests that relationships diagnosed from interannual timescales do have relevance for longer term trends. However, as shown by the distribution of October wind trends (Fig. 6b), the models typically simulate trends that are considerably smaller than the observed. Indeed, <5% of 37-year periods simulate a negative wind trend equal to or larger than observed. As indicated in Supplementary Table 1, while individual models vary somewhat in these characteristics, the simulation of

both weak wind relationships and weak wind trends is true for individual models as well as for the models as a group. This analysis suggests that either the observed period is extremely unique or that the models are deficient in their simulation of relevant processes by which October winds could drive austral fall western Ross Sea ice area trends.

## Discussion

October zonal wind anomalies in the high latitude South Pacific drive a substantial fraction of the western Ross Sea ice variability in the following autumn. The seasonal lag arises because the spring winds drive ice transport anomalies and resulting ice retreat variations that are retained through the summer via enhanced shortwave absorption and ocean heat content anomalies. This relationship has important implications for seasonal forecasts of sea ice, indicating that if October wind variations are known then sea ice in the following March and April in this region is predictable.

Previous work has indicated that trends in autumn western Ross Sea ice cover have little attribution to many modes of variability in the same season[6]. Our results suggest that atmospheric variations during the previous spring season could account for an important component of the sea ice trends in the western Ross Sea during autumn. This indicates that seasonal lags should be considered when diagnosing possible factors driving changes in Antarctic sea ice.

There is little indication that the relevant October wind trends that have been observed are a consequence of anthropogenic

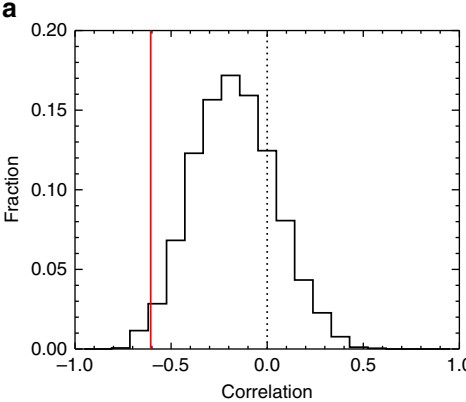

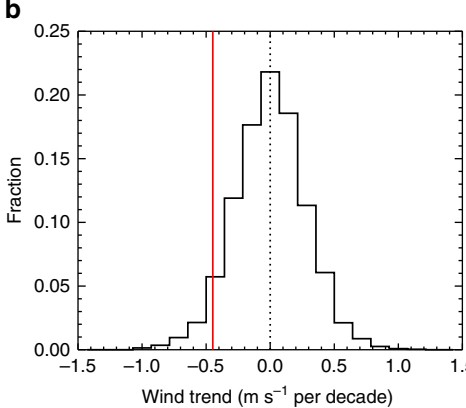

**Fig. 6** Analysis of CMIP5 simulations. **a** Distribution of the correlation coefficients of averaged October zonal wind and the following western Ross Sea March ice area and **b** the distribution of trends in the averaged October zonal wind. The observed values are shown by the *red line*. Analysis is performed on all possible 37-year segments of the model simulations for a direct comparison with the 37-year (1979–2015) observed timeseries. For the analysis shown here, October zonal winds are averaged over the region shown by the *black box* on Fig. 4b

forcing. For example, stratospheric ozone loss has a significant impact on the surface climate during austral summer but not in October[25]. Results from CMIP5 models also show no significant ensemble mean trends in springtime Antarctic atmospheric circulation for 1979–2009[26]. This suggests that the observed October multi-decadal circulation variations arise from natural variability. Analysis from twenty climate models indicates that they typically simulate weak October zonal wind trends in the region of interest when compared to observations. This suggests that the models are deficient in their simulated low-frequency atmospheric circulation variability. Models also typically simulate weak relationships between October zonal winds and the following austral fall sea ice in the western Ross Sea. This, in combination with the weak wind trends, may partly explain why models fail to simulate sea ice trend magnitudes that are comparable to the observed in this region.

## Methods

**Observationally based data sets**. We use a number of data sets to assess relationships of various sea ice conditions with atmospheric variables. This study uses the ERA-Interim atmospheric reanalysis[28] for 1979–2015 for the assessment of surface zonal winds at 10 m height, meridional winds at 10 m height, sea level pressure, and the net surface shortwave flux. As shown in the supplementary material, zonal 10 m wind trends are also assessed from the JRA-55 reanalysis[29] and the MERRA reanalysis[30] for comparison.

We assess the relationship of wind variations to a number of sea ice conditions, including ice concentration, ice motion, and the timing of ice retreat and advance

for 1979–2015. This makes use of the SMMR/SSMI Bootstrap sea ice concentration[31], the Polar Pathfinder sea ice motion[23], and ice advance and retreat timing information based on the passive microwave satellite data[14]. We also assess SST data from NOAA's Optimum Interpolation Sea Surface temperature, version 2 (OISSTv2)[32], which is available from 1982 to 2015.

**Model simulations**. Climate model analysis uses output from pre-industrial control simulations that have been included in the Coupled Model Inter-comparison Project version 5 (CMIP5)[33]. Twenty different climate models are considered as shown in Supplementary Table 1.

Western Ross Sea ice area is defined as the ice area between 150 and 200° E.

**Data availability**. The data that support the findings from this study are available in a number of public repositories. ERA-Interim data are available from http://apps.ecmwf.int/datasets/. The sea ice data are available from the National Snow and Ice Data Center[23, 31] (http://nsidc.org/). NOAA's OISSTv2 is available from https://www.esrl.noaa.gov/psd/data/gridded/data.noaa.oisst.v2.highres.html. Data from CMIP5 models is available from http://cmip-pcmdi.llnl.gov/cmip5/. Information on various data sets was obtained from the Climate Data Guide[34].

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

## Acknowledgements

M.M.H. and L.L. acknowledge support from a grant from the NSF FESD 1338814, NSF 1417642, and NASA 1048926. S.S. acknowledges support from NSF ANT-1440435 and NSF ANT-1341606. We acknowledge the World Climate Research Programme's Working Group on Coupled Modeling, which is responsible for CMIP, and we thank the climate modeling groups (listed in Supplementary Table 1 of this paper) for producing and making available their model output. For CMIP the U.S. Department of Energy's Program for Climate Model Diagnosis and Intercomparison provides coordinating support and led development of software infrastructure in partnership with the Global Organization for Earth System Science Portals.

## Author contributions

M.M.H.: Directed this work and contributions from all authors. M.M.H. and L.L.: Performed the analysis. M.M.H., L.L., M.R. and S.S.: Discussed the results and contributed to writing the manuscript.

## Additional information

**Competing interests:** The authors declare no competing financial interests.

