## [Peer Review File · Nature Communications]

Reviewers' comments:

Reviewer #1 (Remarks to the Author):

Springtime Winds Drive Ross Sea Ice Variability and Change in the Following Autumn

by M. Holland, L. Landrum, M. Raphael, S. Stammerjohn

Review by F. Massonnet, 21 Feb. 2017.

This paper exhibits a seasonal relationship between spring zonal wind anomalies in the Eastern Ross Sea and subsequent fall sea ice area anomalies in the Western Ross Sea. The physical mechanisms underlying this link are also discussed, based on observational datasets and on an atmospheric reanalysis. More intense zonal winds are found to drive dynamical thinning of sea ice, which allows more shortwave radiation to be absorbed during summer and therefore delays sea ice advance in fall. The proposed mechanism is suggested to partly explain the positive trends in sea ice area in this region, which controls a significant part of the circumpolar Antarctic sea ice variability and trends.

I find this study very interesting, novel and worth publishing in Nature Communications up to a few modifications and additional diagnostics listed below. Based on the "Nature Journals guidelines for peer-review" (http://www.nature.com/authors/policies/peer_review.html), I make the following general comments:

"Nature journals receive many more submissions than they can publish. Therefore, we ask peer-reviewers to keep in mind that every paper that is accepted means that another good paper must be rejected. To be published in a Nature journal, a paper should meet four general criteria:

1. Provides strong evidence for its conclusions.

>> There is to me sufficient evidence that the relationship highlighted by the authors is robust. The co-variability between zonal wind anomalies in the Eastern Ross Sea sector in spring and the sea ice area anomalies during the next fall is clear, and it is supported by a sound physical mechanism detailed in the text. As explained below, I would appreciate if the authors provide two additional diagnostics that would "close the loop" of the argument.

2. Novel (we do not consider meeting report abstracts and preprints on community servers to compromise novelty).

>> The paper is novel, in the sense that there is to my knowledge no other work that has showed such a clear seasonal relationship emerging at the seasonal scale. Potential predictability studies using models did point towards this possibility (through other mechanisms though), but the real novelty of this study resides in its use of observational data sets to demonstrate the existence of a dynamical link.

3. Of extreme importance to scientists in the specific field.

>> The paper is relevant for polar scientists, because it highlights that the predictability mechanisms in the Southern Ocean might be more persistent than initially thought. By restricting itself to a specific sector of the Southern Ocean (Ross Sea) the paper also emphasizes once more that sea ice in the Southern Ocean is subject to complex regional dynamics.

4. Ideally, interesting to researchers in other related disciplines.

>> The study might be relevant for researchers from other disciplines or even other other communities (stakeholders). Antarctic sea ice, because of its seasonality, has long been thought to be unpredictable. The authors bring elements suggesting that there might be, in a few regions at

least, potential to forecast large-scale parameters such as sea ice area. This could be relevant for emerging areas such as seasonal sea ice prediction.

I have to add that the authors have made an excellent job in citing the relevant literature on the topic, and in explaining how their study is complementary to the current body of knowledge. There may be one paper that would be worth citing in order to be complete:

Simpkins et al., *J. Clim.*, 2012 (<http://journals.ametsoc.org/doi/pdf/10.1175/JCLI-D-11-00367.1>), as a quite comprehensive review on the seasonal links between atmospheric variability and sea ice variability.

To me, the manuscript is in good shape to be published but there are two additional diagnostics that would make its conclusions even more robust and would strengthen the analyses.

1. The concept of "dynamical thinning" (increased zonal winds in the Eastern Ross Sea mechanically decrease sea ice thickness in the Western Ross Sea) might be not easy to understand for the average reader. I would suggest to add a panel in Fig. 2 showing the regression of October sea ice divergence diagnosed from the ice motion data, on October zonal wind anomalies. It looks like the authors have already made this diagnostic (line 127-129: "Using ice divergence diagnosed from the ice motion data, we find that stronger zonal winds are highly related to enhanced dynamical ice loss from the Western Ross Sea, with a correlation of $R = -0.62$ "), but a map would be much more informative. Divergence is the key diagnostic to make the link between the dynamic part of the argument (wind variability drives sea ice concentration variability in spring) and the thermodynamic part (ice concentration variability in spring drives variability in shortwave absorption by the ocean and hence the date of advance in fall).

2. The average reader may have hard times to figure out why March area anomalies in the Western Ross Sea do not correlate significantly with any of the atmospheric indices in March - they might just not understand where the spring predictability signal has gone. If I understand correctly the reasoning of the authors, the predictability coming from zonal winds anomalies is stored in the ocean as extra heat through enhanced short wave radiation absorption, and re-emerges in March. The authors briefly explain this part of the mechanism (line 138: "This results in anomalously warm sea surface temperatures at the initiation of the ice advance season..."), but this statement is not supported by any material in the paper. I think the readers might be even more convinced by the mechanism if the authors could show how the oceanic heat content anomalies in March, February, January relates to the March ice area. Given the lack of observational data, I would suggest to look for this relationship in an oceanic reanalysis or in a ocean-sea ice stand-alone model integration. I'm sure that this zero-order relationship between upper ocean heat content or SSTs and sea ice area anomalies will be obvious, but I think it's worth showing as an additional figure (or maybe as a panel c) in Fig. 3) to close the loop of the argument.

Other comments:

- The predictand used throughout the manuscript is the March sea ice area in the Western Ross Sea (150-200E). I would be curious to know how sea ice concentration anomalies in March are linked to the preceding October zonal wind anomalies in the $R < -0.6$ domain of Fig. 1d. Could the authors provide such a map in the Supplementary Material? Given what is provided in the current version of the manuscript, I can't figure out if the dynamical thinning induced by increased zonal winds will affect the sea ice pack at all latitudes. From Fig. 2, I anticipate that this relationship will affect mostly sea ice close to the coast, but I would like to have a confirmation of this.

- Like the authors, I am skeptical about the extrapolation made between the mechanism at play at the seasonal scale, which I think is robust, and the long-term trend attribution. The fact that October zonal wind variability in the red box of Fig. 4b hardly explains Western Ross Sea ice trends (while it explains well the year-to-year variability) is to me a clear illustration that the conclusions

drawn from seasonal relationships have to be applied with extreme caution at longer time scales. In fact, do the authors think that the same physical processes as those discussed in the first part of the manuscript are at play for the trends? If so, why is there a westward shift in the zone that best explains the link between winds and sea ice (i.e., from the red to the black arc in Fig. 4d) when the time scale considered for the analysis increases? Saying a few sentences about this shift would be welcome.

- line 59-60: "The presence of seasonal memory in Antarctic sea ice conditions has also been documented in the context of interannual variability". This sentence might appear puzzling to an ordinary reader, because sea ice is almost entirely seasonal. I think that what the authors mean is: "The presence of seasonal memory associated with Antarctic sea ice-ocean interactions has also been...". The physical mechanisms bringing predictability are associated to the storage of heat anomalies in the ocean in between the seasons.

- line 70: "determinant"? Do you mean "predictor"? "Determinant" exists in a dictionary but is used in the context of matrix theory.

- The baseline period for assessment in this paper is 1979-2012. Is there a reason for not considering the recent years during which Antarctic sea ice extent variability was very puzzling? Were the authors limited by the availability of observational data? Could the diagnostics be updated to go at least to 2015?

- Fig. 1: in the caption, replace "October" by "preceding October".

- Fig. 2: it would be helpful to plot the contour of correlation $< - 0.6$ from Fig. 1d on the map.

- Fig. 3: Please specify the baseline period (I assume 1979-2012?). Also, would it be possible to communicate the significance of the regression by e.g. masking points where the correlations between net short-wave anomalies and the October zonal wind anomalies are not significant?

Reviewer #2 (Remarks to the Author):

Using observational data products, Holland et al. clearly demonstrate a correlation between interannual zonal wind anomalies in October and subsequent sea ice area anomalies in the Ross Sea sector in March.

The study furthers our understanding of the factors that contribute to sea ice variability in the Ross Sea region and will likely help scientists to better diagnose GCM biases that may be contributing to the fact that GCMs do not simulate the observed negative trend in Antarctic sea ice, particularly in the Ross Sea region.

I recommend this study for publication after a few issues have been addressed.

First, I am wondering why the authors only examine data up to 2012. There are 4 additional years of ERA-I reanalysis available. Also, what are surface winds? Are these the 10-m ERA-I winds? This could be clarified in the Methods section.

I appreciate the clarity and simplicity of this study; however, in some ways I find it overly simple. The authors sometimes ask the reader to fill in the gaps. For example, it would be nice to see figure panels of SST and Feb/Mar timing of sea ice advance regressed on the October zonal wind index to support the authors' claims, rather than taking their word for it that enhanced downward shortwave anomalies lead to increased SST that leads to delayed sea ice advance in fall.

The authors also never mention statistical significance of correlations, regressions or trends. I appreciate that most of the correlations quoted in the text are quite high and, therefore, likely highly significant. However, the figures could include simple hatching (or other plotting techniques) to illustrate statistical significance.

Line 117: What is consistent with Hosking et al? Hosking et al do not look at lagged-correlations from what I recall. Are the authors referring to the fact that they show that anomalies in ASL longitude, etc. also have an relationship to sea ice anomalies?

Regarding the trends, I am wondering whether the trends in zonal wind are robust across different reanalyses, such as MERRA or JRA, and robust to extending the analysis to year 2016.

This study stands alone; however, I feel that some model assessment would enhance the paper. The authors mention many times the mismatch between observations and models. Would it not be fairly straight forward to examine whether the models show this detrended correlation between October U and March Ross Sea sea ice? This would be easier to do if the authors averaged Oct U over a box rather than some "region of high correlation" (a box is also more reproducible and not sensitive to the length of the time series). Are the models able to get this interannual relationship at all? It would be useful to know whether or not this interannual relationship is the way in which models are biased or whether it is something else. If they do get this interannual, relationship, then a similar trend analysis could be done with the models as was done for the observations. I assume the models don't get the observed negative U trend in the Amundsen and Ross Sea regions.

Finally, I find it interesting that the relationship between March Ross Sea sea ice and the October ASL is opposite to what has been discussed in the Antarctic sea ice literature for contemporaneous correlations (this is also quite evident in Supplementary figure 3). I think that this could be emphasized to highlight the complex coupling between Ross Sea sea ice and the atmospheric circulation.

Reviewers' comments:

Reviewer #1 (Remarks to the Author):

I find this study very interesting, novel and worth publishing in Nature Communications up to a few modifications and additional diagnostics listed below. Based on the "Nature Journals guidelines for peer-review", I make the following general comments:

"Nature journals receive many more submissions than they can publish. Therefore, we ask peer-reviewers to keep in mind that every paper that is accepted means that another good paper must be rejected. To be published in a Nature journal, a paper should meet four general criteria:

1. Provides strong evidence for its conclusions.

>> There is to me sufficient evidence that the relationship highlighted by the authors is robust. The co-variability between zonal wind anomalies in the Eastern Ross Sea sector in spring and the sea ice area anomalies during the next fall is clear, and it is supported by a sound physical mechanism detailed in the text. As explained below, I would appreciate if the authors provide two additional diagnostics that would "close the loop" of the argument.

Please see further information on these diagnostics below.

2. Novel (we do not consider meeting report abstracts and preprints on community servers to compromise novelty).

>> The paper is novel, in the sense that there is to my knowledge no other work that has showed such a clear seasonal relationship emerging at the seasonal scale. Potential predictability studies using models did point towards this possibility (through other mechanisms though), but the real novelty of this study resides in its use of observational data sets to demonstrate the existence of a dynamical link.

Thanks.

3. Of extreme importance to scientists in the specific field.

>> The paper is relevant for polar scientists, because it highlights that the predictability mechanisms in the Southern Ocean might be more persistent than initially thought. By restricting itself to a specific sector of the Southern Ocean (Ross Sea) the paper also emphasizes once more that sea ice in the Southern Ocean is subject to complex regional dynamics.

4. Ideally, interesting to researchers in other related disciplines.

>> The study might be relevant for researchers from other disciplines or even other

other communities (stakeholders). Antarctic sea ice, because of its seasonality, has long been thought to be unpredictable. The authors bring elements suggesting that there might be, in a few regions at least, potential to forecast large-scale parameters such as sea ice area. This could be relevant for emerging areas such as seasonal sea ice prediction.

I have to add that the authors have made an excellent job in citing the relevant literature on the topic, and in explaining how their study is complementary to the current body of knowledge. There may be one paper that would be worth citing in order to be complete:

Simpkins et al., J. Clim., 2012 (<http://journals.ametsoc.org/doi/pdf/10.1175/JCLI-D-11-00367.1>), as a quite comprehensive review on the seasonal links between atmospheric variability and sea ice variability.

Thanks for this suggestion. This is indeed a relevant publication, which discusses the contemporaneous relationships between SAM/ENSO and sea ice. We now include a citation to it.

To me, the manuscript is in good shape to be published but there are two additional diagnostics that would make its conclusions even more robust and would strengthen the analyses.

1. The concept of "dynamical thinning" (increased zonal winds in the Eastern Ross Sea mechanically decrease sea ice thickness in the Western Ross Sea) might be not easy to understand for the average reader. I would suggest to add a panel in Fig. 2 showing the regression of October sea ice divergence diagnosed from the ice motion data, on October zonal wind anomalies. It looks like the authors have already made this diagnostic (line 127-129: "Using ice divergence diagnosed from the ice motion data, we find that stronger zonal winds are highly related to enhanced dynamical ice loss from the Western Ross Sea, with a correlation of $R = -0.62$ "), but a map would be much more informative. Divergence is the key diagnostic to make the link between the dynamic part of the argument (wind variability drives sea ice concentration variability in spring) and the thermodynamic part (ice concentration variability in spring drives variability in shortwave absorption by the ocean and hence the date of advance in fall).

The important quantity for the dynamical thinning is the net divergence of ice from the Western Ross Sea domain, which is equivalent to the net ice area flux through the boundary of that domain. As such, a map of the grid-cell level divergence is not very illustrative. To better communicate this, we have revised the wording here to more clearly articulate that a significant correlation is found with the total ice area transport from the western Ross Sea domain.

2. The average reader may have hard times to figure out why March area anomalies

in the Western Ross Sea do not correlate significantly with any of the atmospheric indices in March - they might just not understand where the spring predictability signal has gone. If I understand correctly the reasoning of the authors, the predictability coming from zonal winds anomalies is stored in the ocean as extra heat through enhanced short wave radiation absorption, and re-emerges in March. The authors briefly explain this part of the mechanism (line 138: "This results in anomalously warm sea surface temperatures at the initiation of the ice advance season..."), but this statement is not supported by any material in the paper. I think the readers might be even more convinced by the mechanism if the authors could show how the oceanic heat content anomalies in March, February, January relates to the March ice area. Given the lack of observational data, I would suggest to look for this relationship in an oceanic reanalysis or in a ocean-sea ice stand-alone model integration. I'm sure that this zero-order relationship between upper ocean heat content or SSTs and sea ice area anomalies will be obvious, but I think it's worth showing as an additional figure (or maybe as a panel c) in Fig. 3) to close the loop of the argument.

Thanks for this suggestion. We now include analysis of the relationship of zonal wind anomalies to SST variability. This includes additional discussion on lines 157-160 and an additional figure (Figure 3b) showing the regression of monthly sea surface temperature from NOAA's Optimum Interpolation Sea Surface temperature version 2 (OISSTv2) on the October zonal wind anomalies.

Other comments:

- The predictand used throughout the manuscript is the March sea ice area in the Western Ross Sea (150-200E). I would be curious to know how sea ice concentration anomalies in March are linked to the preceding October zonal wind anomalies in the $R < -0.6$ domain of Fig. 1d. Could the authors provide such a map in the Supplementary Material? Given what is provided in the current version of the manuscript, I can't figure out if the dynamical thinning induced by increased zonal winds will affect the sea ice pack at all latitudes. From Fig. 2, I anticipate that this relationship will affect mostly sea ice close to the coast, but I would like to have a confirmation of this.

We have now provided a correlation map of the March sea ice conditions with the October zonal winds in the supplemental material (Supplementary Fig. 1) and refer to this on lines 107-109 (bottom of page 5) in the text. It shows significant correlations of March sea ice concentration across the western Ross Sea domain.

- Like the authors, I am skeptical about the extrapolation made between the mechanism at play at the seasonal scale, which I think is robust, and the long-term trend attribution. The fact that October zonal wind variability in the red box of Fig. 4b hardly explains Western Ross Sea ice trends (while it explains well the year-to-

year variability) is to me a clear illustration that the conclusions drawn from seasonal relationships have to be applied with extreme caution at longer time scales. In fact, do the authors think that the same physical processes as those discussed in the first part of the manuscript are at play for the trends? If so, why is there a westward shift in the zone that best explains the link between winds and sea ice (i.e., from the red to the black are in Fig. 4d) when the time scale considered for the analysis increases? Saying a few sentences about this shift would be welcome.

As articulated in the paper, we agree that caution is needed when extrapolating from interannual relationships to long term trends.

The trend in the winds is sizable for a region that is shifted westward from the region of highest seasonal correlations. Note that this is still a region of high interannual correlation (as shown on Figure 4c) and there is overlap between the two regions. These winds do have a high correlation with the Western Ross Sea ice area. However, as noted in the text at the top of page 9 (lines 183-185) and shown in Supplemental Fig 1b, the winds in the high trend region “are most influential in the western part of the sea ice analysis domain.” Notably, and as mentioned in the text, trends in the ice are also largest in the western part of the domain.

To address the question of whether similar physical processes are at play, we have also now assessed the interannual relationships of other variables (ice motion, ice area divergence, net shortwave, etc.) with winds in the high trend region (black box on Fig 4d). They are quite similar to the analysis shown in Figs 1-3. We now mention this in the revised manuscript (lines 186-189). Because of this, we believe that a similar mechanism could explain wind-driven ice trends and the interannual variability.

Finally, following a request from Reviewer #2 and the journal editor, we have added analysis of climate model simulations to the manuscript. Because of this, we have modified the wind region for the trend analysis to be a standard latitude-longitude box (black region on Fig. 4d) for comparison with the climate models. This gives very similar results to the region used in the original manuscript.

- line 59-60: "The presence of seasonal memory in Antarctic sea ice conditions has also been documented in the context of interannual variability". This sentence might appear puzzling to an ordinary reader, because sea ice is almost entirely seasonal. I think that what the authors mean is: "The presence of seasonal memory associated with Antarctic sea ice-ocean interactions has also been...". The physical mechanisms bringing predictability are associated to the storage of heat anomalies in the ocean in between the seasons.

Changed as suggested.

- line 70: "determinant"? Do you mean "predictor"? "Determinant" exists in a dictionary but is used in the context of matrix theory.

Yes. We mean "predictor" and have changed the wording here.

- The baseline period for assessment in this paper is 1979-2012. Is there a reason for not considering the recent years during which Antarctic sea ice extent variability was very puzzling? Were the authors limited by the availability of observational data? Could the diagnostics be updated to go at least to 2015?

We have now updated the analysis through 2015.

- Fig. 1: in the caption, replace "October" by "preceding October".

Changed as suggested.

- Fig. 2: it would be helpful to plot the contour of correlation < -0.6 from Fig. 1d on the map.

This contour is now shown on Figure 2.

- Fig. 3: Please specify the baseline period (I assume 1979-2012?). Also, would it be possible to communicate the significance of the regression by e.g. masking points where the correlations between net short-wave anomalies and the October zonal wind anomalies are not significant?

We now include the period of analysis (1979-2015) in the figure caption.

Reviewer #2 (Remarks to the Author):

Using observational data products, Holland et al. clearly demonstrate a correlation between interannual zonal wind anomalies in October and subsequent sea ice area anomalies in the Ross Sea sector in March.

The study furthers our understanding of the factors that contribute to sea ice variability in the Ross Sea region and will likely help scientists to better diagnose GCM biases that may be contributing to the fact that GCMs do not simulate the observed negative trend in Antarctic sea ice, particularly in the Ross Sea region.

I recommend this study for publication after a few issues have been addressed.

First, I am wondering why the authors only examine data up to 2012. There are 4 additional years of ERA-I reanalysis available. Also, what are surface winds? Are these the 10-m ERA-I winds? This could be clarified in the Methods section.

We have now updated analysis through 2015. This is the time period for which all the datasets used are available. We have also clarified in the methods section that the “surface” winds are indeed the 10-m winds.

I appreciate the clarity and simplicity of this study; however, in some ways I find it overly simple. The authors sometimes ask the reader to fill in the gaps. For example, it would be nice to see figure panels of SST and Feb/Mar timing of sea ice advance regressed on the October zonal wind index to support the authors' claims, rather than taking their word for it that enhanced downward shortwave anomalies lead to increased SST that leads to delayed sea ice advance in fall.

Thank you for this suggestion. To fill in the gaps, we now show analysis of both SST and ice advance timing. More specifically, we show the regression of the timing of ice advance as well as the net surface shortwave radiation on Figure 3a. The regression of SST on the zonal wind anomalies is now shown in Figure 3b.

The authors also never mention statistical significance of correlations, regressions or trends. I appreciate that most of the correlations quoted in the text are quite high and, therefore, likely highly significant. However, the figures could include simple hatching (or other plotting techniques) to illustrate statistical significance.

We now indicate statistical significance on the figures.

Line 117: What is consistent with Hosking et al? Hosking et al do not look at lagged-correlations from what I recall. Are the authors referring to the fact that they show that anomalies in ASL longitude, etc. also have an relationship to sea ice anomalies?

Sorry for the confusing wording here. We meant to state that the possibility of the location of the ASL being important for ice variability is consistent with Hosking et al. We have now restated this as: “This is generally consistent with other work assessing concurrent relationships that has indicated that not only the depth of the ASL but its location can affect relationships to sea ice.” (line 132-134 in revised manuscript).

Regarding the trends, I am wondering whether the trends in zonal wind are robust across different reanalyses, such as MERRA or JRA, and robust to extending the analysis to year 2016.

We have extended the analysis through 2015 (which is the end year for which all datasets are available). The trends from ERA-I are indeed robust to this year. We have also assessed trends in MERRA and JRA55 October winds. While details differ somewhat, all three reanalysis products do show a broad weakening of the winds in the Ross Sea region. We mention this finding within the text on line 179-180.

This study stands alone; however, I feel that some model assessment would enhance the paper. The authors mention many times the mismatch between observations and models. Would it not be fairly straight forward to examine whether the models show this detrended correlation between October U and March Ross Sea sea ice? This would be easier to do if the authors averaged Oct U over a box rather than some "region of high correlation" (a box is also more reproducible and not sensitive to the length of the time series). Are the models able to get this interannual relationship at all? It would be useful to know whether or not this interannual relationship is the way in which models are biased or whether it is something else. If they do get this interannual, relationship, then a similar trend analysis could be done with the models as was done for the observations. I assume the models don't get the observed negative U trend in the Amundsen and Ross Sea regions.

We have now incorporated analysis from twenty different CMIP5 models into the study. In particular, we analyze multi-century pre-industrial control simulations from the models (resulting in over 11,000 years of model integration). From these simulations, we compute the correlation between October zonal winds and Western Ross Sea March ice area for all possible 37-year timeseries from the models. We also compute all possible 37-year October zonal wind trends. For this analysis, the winds are averaged over a box as also now used for the observational analysis of trends (black box on Figure 4b, and analysis shown by black lines on Fig. 4c and 4d). Distributions that summarize the model results are now shown in Figure 5a and 5b.

This analysis illustrates that, while the models do tend to simulate Western Ross Sea March ice area loss following strong zonal October zonal winds (and have mostly a negative correlations on Fig 5a), they very rarely simulate a correlation that is as strong as the one seen in observations. The models also very rarely simulate zonal wind trends of the magnitude that are observed. This suggests that the models are deficient in both their simulation of the wind-ice interactions and their simulation of the wind trends. This may indeed contribute to the lack of strongly increasing sea ice trends in the Western Ross Sea in models. Text describing these results is not included on pages 10-12 and mentioned in the abstract and discussion sections.

Finally, I find it interesting that the relationship between March Ross Sea sea ice and the October ASL is opposite to what has been discussed in the Antarctic sea ice literature for contemporaneous correlations (this is also quite evident in Supplementary figure 3). I think that this could be emphasized to highlight the complex coupling between Ross Sea sea ice and the atmospheric circulation.

Thanks for this suggestion. We agree that the different ASL influence on seasonal lagged versus contemporaneous timescales is interesting and have now highlighted this in the text: "Interestingly, the relationship between March sea ice area in the western Ross Sea and the depth of the ASL is of

opposite sign for the seasonal lagged relationships discussed here compared to a contemporaneous relationship.” We also now cite Turner et al. (2009) as a reference for the contemporaneous relationship.

REVIEWERS' COMMENTS:

Reviewer #1 (Remarks to the Author):

Second review of "Springtime Winds Drive Ross Sea Ice Variability and Change in the Following Autumn" by M. M. Holland et al.

by François Massonnet

The authors have addressed my concerns and comments appropriately. I'm also glad to see the additional analysis about the CMIP5 ensemble, which features systematic deficiencies in the ability of climate models to simulate the magnitude of the relationship identified in the observational datasets. I'm sure that this finding will lead to further investigations of high importance to understand the origin of model biases in the Southern Ocean.

Two final comments:

- Given the large audience that this article is expected to reach, I would somewhere display a climatology of winds in the Antarctic region (for example overlaid on Fig. 1d). The paper goes very quickly in discussions about anomalies in Spring zonal winds, but it would be good to know the mean state they deviate from.

- line 127: associate  associated.

Reviewer #2 (Remarks to the Author):

Thank you to the authors for considering my comments and including some of my suggestions in the revised manuscript. The additions address many of my questions and likely questions that other readers may have had. I believe that this is an important contribution and will help to further our understanding of Antarctic sea ice variability and trends. I recommend this manuscript for publication in Nature Communications.

The only comment that I have is regarding lines 272-273. There is no reference for this statement and one should be added. I agree with the authors, based on the MMM SAM trends from the CMIP5 models, which are positive in the SON season, but this statement needs to be supported by a specific study (e.g. Swart et al. 2015). A reference is also necessary because this statement is assumed to be true, but not explicitly stated as such, in the GCM analysis. The fact that the authors chose to examine the Pi-Control simulations (rather than the historical simulations) to evaluate the distribution of Oct U trends assumes that the observed Oct U trend has no anthropogenic component. I think that this is probably fine, but the authors need to explain their reasoning for using the Pi-Control simulations a bit more clearly.

Note: The Swart et al. (2015) study cautions against using reanalysis winds to examine the impact of wind on the S. Ocean. This study should be cited somewhere. Their analysis uses the cross-calibrated multiplatform (CCMP) ocean surface wind vector analyses of Atlas et al. (2011) for the period 1988–2011 to compare with reanalyses. I am not familiar with this data product. Have the authors looked at the CCMP data at all? I am curious to know how the Oct U trends compare with ERA-I.

- Karen Smith

Response to reviewers

We thank the reviewers for the additional comments on our manuscript. Below is our response to these comments in bold with the reviewer's comments in plain text.

REVIEWERS' COMMENTS:

Reviewer #1 (Remarks to the Author):

Second review of "Springtime Winds Drive Ross Sea Ice Variability and Change in the Following Autumn" by M. M. Holland et al.

by François Massonnet

The authors have addressed my concerns and comments appropriately. I'm also glad to see the additional analysis about the CMIP5 ensemble, which features systematic deficiencies in the ability of climate models to simulate the magnitude of the relationship identified in the observational datasets. I'm sure that this finding will lead to further investigations of high importance to understand the origin of model biases in the Southern Ocean.

Two final comments:

- Given the large audience that this article is expected to reach, I would somewhere display a climatology of winds in the Antarctic region (for example overlaid on Fig. 1d). The paper goes very quickly in discussions about anomalies in Spring zonal winds, but it would be good to know the mean state they deviate from.

Thanks for this good suggestion. We tried including this on Fig. 1d but it made the figure too busy. Instead, we now include a new figure (Fig. 2) which displays the climatological winds in October.

- line 127: associate  associated.

Changed

Reviewer #2 (Remarks to the Author):

Thank you to the authors for considering my comments and including some of my suggestions in the revised manuscript. The additions address many of my questions and likely questions that other readers may have had. I believe that this is an important contribution and will help to further our understanding of Antarctic sea ice variability and trends. I recommend this manuscript for publication in Nature Communications.

The only comment that I have is regarding lines 272-273. There is no reference for this statement and one should be added. I agree with the authors, based on the MMM SAM trends from the CMIP5 models, which are positive in the SON season, but this statement needs to be supported by a specific study (e.g. Swart et al. 2015). A reference is also necessary because this statement is assumed to be true, but not explicitly stated as such, in the GCM analysis. The fact that the authors chose to examine the Pi-Control simulations (rather than the historical simulations) to evaluate the distribution of Oct U trends assumes that the observed Oct U trend has no anthropogenic component. I think that this is probably fine, but the authors need to explain their reasoning for using the Pi-Control simulations a bit more clearly.

Thanks for this suggestion. We have added additional information for the text on lines 272-273 (now 313-316 in the revised manuscript) as follows:
“There is little indication that the relevant October wind trends that have been observed are a consequence of anthropogenic forcing. For example, stratospheric ozone loss has a significant impact on the surface climate during austral summer but not in October²⁵. Results from CMIP5 models also show no significant ensemble mean trends in springtime Antarctic atmospheric circulation for 1979-2009²⁶.”

And have added references here to Thompson et al., 2011 and Swart et al., 2015.

We also now explicitly note on lines 247-250 in the revised manuscript that:
“We use output from pre-industrial control simulations given that there is little evidence that anthropogenic forcing from ozone loss or greenhouse gases are driving Antarctic wind changes in spring^{25, 26}.” (with a reference to the same publications)

Note: The Swart et al. (2015) study cautions against using reanalysis winds to examine the impact of wind on the S. Ocean. This study should be cited somewhere. Their analysis uses the cross-calibrated multiplatform (CCMP) ocean surface wind vector analyses of Atlas et al. (2011) for the period 1988–2011 to compare with reanalyses. I am not familiar with this data product. Have the authors looked at the CCMP data at all? I am curious to know how the Oct U trends compare with ERA-I.

As mentioned above, we now cite the Swart et al. (2015) paper. Additionally, we believe that the ERA-I winds are reasonable to use for our study given that, as shown in the manuscript, they exhibit significant physically plausible relationships to a number of independent datasets.

We have not assessed the CCMP data. CCMP uses ERA-Interim for a first guess and the satellite winds used in the CCMP product are not available over sea ice. Given that the October winds we assess occur in part over ice covered regions, we are uncertain if it would be an appropriate product for the analysis performed here.

- Karen Smith